# Reproductive Technologies Used in Female Neo-Tropical Hystricomorphic Rodents

**DOI:** 10.3390/ani12050618

**Published:** 2022-03-01

**Authors:** Kavita Ranjeeta Lall, Kegan Romelle Jones, Gary Wayne Garcia

**Affiliations:** 1Department of Food Production (DFP), Faculty of Food and Agriculture (FFA), University of the West Indies (UWI), St. Augustine Campus, St. Augustine 999183, Trinidad and Tobago; k_lee_24@yahoo.com (K.R.L.); prof.gary.garcia@gmail.com (G.W.G.); 2Department of Basic Veterinary Sciences (DBVS), School of Veterinary Medicine (SVM), Faculty of Medical Sciences, University of the West Indies (UWI), St. Augustine Campus, St. Augustine 999183, Trinidad and Tobago

**Keywords:** agouti, *Dasyprocta leporina*, lappe, *Cuniculus paca*, capybara, *Hydrochoerus hydrochaeris*, reproductive tract

## Abstract

**Simple Summary:**

This literature review focused on the reproductive technologies used in female neo-tropical hystricomorphic rodents. Reproductive technologies aid in efficient reproduction, which is important in these species as they are hunted and valued for their meat. Knowledge of the anatomy and physiology would aid in assisted reproductive techniques, thus attention was given to these areas. Within this group of rodent species there were similar characteristics, some of which have been highlighted as well as any unique features. Some reproductive technologies used included colpocytology, ultrasonography, and hormonal analysis.

**Abstract:**

This is the second of two literature reviews that focuses on the female reproductive anatomy and reproductive technologies used in selected neo-tropical hystricomorphic rodents. The rodents chosen included the agouti (*Dasyprocta leporina*), the capybara (*Hydrochoerus hydrochaeris*), and the paca (*Cuniculus paca*). Over seventy references were used, spanning the years 1919–2021. Knowledge of the reproductive tract is important in understanding any unique features, which may affect the use of reproductive technologies. Some unique characteristics common to these species included the presence of a vaginal closure membrane and a lobulated placenta with a vascular sub-placenta. The capybara had hyperpigmentation of the vagina that was unique to each individual, while the agouti and paca had accessory corpora lutea, in addition to the main one. Some reproductive technologies have been used, with attempts at estrous induction and synchronization taking place within the past five years. Even though most work has been done over the past twenty years, there is still a dearth of information.

## 1. Introduction

Hystricomorphic rodents belong to the suborder of Hystricomorpha, which refers to one of the four types of rodent skulls based on the nature of their zygomasseteric system, whereby the anterior part of the masseter medialis runs from the medial side of the orbit through an enlarged infraorbital foramen and to the lateral surface of the rostrum and in extreme cases, such as in the capybara (*Hydrochoerus* spp.), its origin extends as far forward as the pre-maxilla [1,2]. 

Neo-tropical hystricomorphic rodents, such as the agouti (*Dasyprocta leporina*), capybara (*Hydrochoerus hydrochaeris*), and paca (*Cuniculus paca*), possess the potential for domestication and are of importance as they serve as game species, in addition to their ecological roles—such as scatter-hoarders [3,4,5,6,7]. Therefore, wildlife farming can help to create a captive-bred stock, which will create a gene pool, aid in food production, create employment, and aid in conservation [7,8].

Optimizing breeding programs entails the use of reproductive technologies in order to modify reproductive performance in such a way as to improve animal reproduction [9,10,11,12]. Improvement in reproduction in these programs is an important aspect of ensuring the survival of species, such as the *Dasyprocta* spp., for conservation and utilization (i.e., as analternative protein source for human consumption), as many factors threaten the future of these species [7]. 

Agoutis (*Dasyprocta* spp.) are medium-sized rodents that weigh between 2–3.5 kg [13,14,15,16,17]. Studies defining the age of puberty in females are lacking, however, Guimarães et al. [18] showed that, under the male effect, puberty in *D. prymnolopha* was attained at nine months. They exhibit a continuous polyestrous cycle [19], with an estrous cycle varying among the different species; 34.2 ± 2.1 days in *D. aguti* [20], 30.69 ± 4.65 days [19] and 29.94 ± 6.77 days [21] in *D. prymnolopha,* and 28.2 ± 0.7 days in *D. leporina*, [22]. These rodents have a gestation period of 103–104 days [19,23], with an average litter size of 1.7–2 precocial young [24,25,26]. Single birth weights were higher than multiple ones with offspring weighing 209–310 g [26,27]. The interval between two young being birthed was approximately one hour, with the female squatting while exhibiting contractions. After expulsion of the fetus, the fetal membranes and placenta were consumed by the mother [28]. 

The capybara (*H. hydrochaeris*) is the largest rodent in the world, weighing 54.1 ± 1.9 kg [29]. They reach puberty between 10–12 months old, attaining sexual maturity at 26 months of age, with an estrous cycle of 7.5 ± 1.2 days [30,31]. They are considered to be spontaneous ovulators [32]. They could breed throughout the year, but some authors described a seasonal peak breeding pattern (seasonal peak in births) with a gestation period of 145–150 days that culminates in an average litter size of 3.3–3.8 [30,31,33]. The young had a birth weight of approximately 1.5–2.0 kg [34]. Infanticide rates increased in group settings with unfamiliar females as opposed to females that birthed in isolation pens and even in groups with familiar females [35]. 

The paca (*C. paca*) is a medium-sized rodent, weighing on average 7.7 kg [36]. In captivity, females were found to be polyestrous animals that bred year-round with puberty being attained at 8–12 months and having an estrous cycle of 32.5 ± 3.7 days and a gestation length of 148.6 ± 4.8 days [37], culminating in the production of one offspring [38]. Newborn pacas weighed 691–902 g [39]. 

The objectives of this review were to document the gross anatomy and histology of the female reproductive systems, in addition to the reproductive technologies reported for the female agouti (*D. leporina*), capybara (*H. hydrochaeris*), and paca (*C. paca*). Knowledge of the basic anatomy and physiology of the reproductive tract is important before attempting to understand and use assisted reproductive techniques [11].

## 2. Methodology

Search engines such as Google Scholar, UWIlinc, and Pubmed were used for exhaustive literature searches. Keywords searched included *Dasyprocta leporina*, *Hydrochoerus hydrochaeris*, *Cuniculus paca*, female reproduction, hystricomorphic rodents, reproductive technology, female reproductive tract, vaginal cytology, estrus induction, and ovaries. There were 200 articles shortlisted from the search, spanning the years 1919–2021, of which 74 were relevant to writing this review. The inclusion criteria included the anatomy and histology of the female reproductive tract for each species and the reproductive technologies, such as estrous cycle identification, estrus induction, estrous synchronization, vaginal cytology, and ovarian tissue preservation.

## 3. Agouti (*Dasyprocta leporina*)

### 3.1. Gross Anatomy of the Female Reproductive Tract

The ovaries were pale yellow in colour with tiny translucent areas that were oval in shape and had a smooth surface with no ovarian bursa [14,20,25,40,41,42]. The ovary, found within adipose tissue and supported by the mesovarium, was enclosed by the tunica albuginea and consisted of two layers—the cortex, which was the outer layer, and the medulla, which was the inner layer [40,41,42]. Weir [20] stated that the ovaries weighed 0.207 ± 0.019 g, however, Oliveira et al. [42] found that both the right and left ovaries had the same weight of 0.09 ± 0.01 g. Further morphometric data of the ovary could be seen in Table 1.

There were three parts of the uterine tubes: the infundibulum, ampulla, and isthmus [25,42,43]. The paired uterine horns increased in width as they approached the uterus and had a mean length of 2.2 ± 0.19 cm, with the left uterine horn being more tortuous [41]. They also noted that the mucosal surface contained folds and branches and the uterine horns were attached to the abdominal wall via the broad ligament. There was also a prominent cranial loop of the fallopian tube [20,41]. Mayor et al. [25] found that the length of the uterine tubes of *D. fuliginosa* was 0.504 ± 0.123 cm and, also, that the length did not differ with the reproductive state. This was in stark contrast to de Oliveira et al. [42], who found that the length of the uterine tubes for the *D. leporina* was 5.023 ± 2.51 cm and 4.911 ± 2.05 cm for the left and right uterine tubes, respectively. They also reported that the left and right uterine horns were 5.052 ± 0.423 cm and 5.176 ± 0.419 cm in length, respectively, which then went on to form a false uterine body caudally that varied in diameter between the different estrous cycle phases.

Singh et al. [41] noted that the duplex uterus had an average length of 1.04 ± 0.55 cm. The uterine body was small, with a utero-tubal junction where the uterine horns met the uterus cranially, while, caudally, the cervix was located. A thick septum divided the uterine body, resulting in two lumens. Weir [20] and Mayor et al. [25] also gave a similar gross description of the uterus. However, de Oliveira et al. [42], described the uterus as being partially double, comprising of two horns and a septate cervix (two internal cervical ostia), in which the septum disappeared caudally to create a single ostium (external cervical ostium) that opened into the vagina. On the other hand, Martins et al. [44] stated that, despite there being only one external cervical ostium, it was characterized as a double uterus.

According to Singh et al. [41], the cervix had a mean length of 2.15 ± 2.10 cm and was located at the pelvic inlet, which was dorsal to the urinary bladder and protruded into the proximal vaginal lumen (fornix). These findings agreed with de Oliveira et al. [42], who gave a similar description and obtained a mean length of 1.701 ± 0.639 cm. It was found to be a muscular, thick-walled structure [25,41]. Mayor et al. [25] also noted the animals had a double cervix.

The vagina was supported by the broad ligament and was located dorsal to the rectum, had a mean length of 7.28 ± 1.14 cm (non-pregnant animals), and contained longitudinal mucosal folds extending throughout the length of the vagina [41]. The anatomical location disagreed with de Oliveira et al. [42] who stated that the vagina was found ventral to the rectum; however, they agreed that it was a long tube, as it measured 5.974 ± 0.206 cm in length with a diameter of 1.096 ± 0.056 cm. In most mammalian species, the vagina is located ventral to the rectum. These authors also reported that the vagina directly opened into the vulva, in the vaginal ostium. Mayor et al. [25] also noted the longitudinal folds, but stated that the vagina was thin-walled and measured (in length) 7.77 ± 2.70 cm in a non-pregnant animal (follicular phase), 7.20 ± 0.71 cm in a non-pregnant animal (luteal phase), and 8.14 ± 1.68 cm in a pregnant animal. Table 2 shows further measurements of the vagina. Weir [20] also noted the presence of a vaginal closure membrane, which was perforated for 1–10 days during estrus. This membrane was also noted by de Oliveira et al. [42] in animals that were in diestrus and also during pregnancy. The vaginal closure membrane, covering the vaginal orifice, could therefore aid in the differentiation of the phases of the estrous cycle as it could be formed shortly after the heat period once copulation did not take place or after the expulsion of the vaginal plug [45]. The vaginal closure membrane was found to persist during pregnancy and up to the first stage of estrous, where it then disintegrated when the female entered into heat, ready for copulation [45].

As stated by Singh et al. [41], the vulva contained no vaginal vestibule. The urinary and genital tracts were also found to be separate from one another. These findings were similar to that of de Oliveira et al. [42], who also noted that the vulvar lips were minimized and that there was no vestibule present. The vulva appeared pale and normal in size during the metestrus and diestrus phases, however, during the estrus phase, it would become reddened and enlarged.

The clitoris was seen as a protruding, conical-shaped structure in the vulvar area that was covered by skin and hair [42]. At the top of the clitoris were two hard, protruding lateral keratinized spicules, between which the urethra, with an opening to the urethral ostium, was located. It should be noted that the female had four pairs of functional mammary glands (cranial teat pair, two sets of abdominal teat pairs, and caudal teat pair) [27].

### 3.2. Histology of the Female Reproductive Tract

The ovaries were found to be covered by simple, cuboidal to squamous epithelium, with the cortex consisting of cellular connective tissue, in which ovarian follicles—at different stages of maturation—and corpora lutea could be found, while the medulla was made up of loose connective tissue that contained blood vessels [14,40,41,42,46]. De Oliveira et al. [42] noted the presence of developing follicles and corpus luteum in regression during the proestrus phase; Graafian follicles were present during estrus; a hemorrhagic corpus was present during metestrus; and a mature corpus luteum was present during diestrus. Almeida et al. [40] also noted that, in pregnant agoutis, there were more corpora lutea than follicles, with two to three larger ones and many smaller ones, while, in non-pregnant agoutis, there were many small corpora lutea. Weir [20] made similar discoveries of numerous cell types, noting that the primary oocytes were found in the cortex; the follicle was surrounded by theca externa cells (predominant; 50–100 µ in thickness) and theca interna cells (scattered throughout the theca externa); accessory corpora lutea were smaller than true corpora lutea; and the ovary contained an abundance of interstitial tissue. According to Mayor et al. [25], the follicles of females in the follicular phase of the estrous cycle had a greater number and diameter of antral follicles than females in the luteal phase and pregnant animals. Morphometric data of the ovarian components can be seen in Table 3.

The uterine horns were lined with simple columnar cells, which decreased in height as they approached the uterus [41,43]. This agreed with the findings by Mayor et al. [25] and de Oliveira et al. [42] who noted that the uterine tubes had a highly folded epithelial mucosa with a ciliated simple columnar epithelium. Further details given by de Oliveira et al. [42] stated that, in addition to the aforementioned mucosal layer, there were other layers, which were also stated by Fortes et al. [43]: The lamina propria, muscular layer, and serosa. Mayor et al. [25] noted that the tunica muscularis consisted of an inner circular layer and outer longitudinal layer of smooth muscle cells and, together with the tunica mucosa, were narrower in the infundibulum. De Oliveira et al. [42] discovered that the infundibulum was comprised of many folds, a thin serous muscular layer (106.81 ± 4 μm), and was lined by simple ciliated columnar epithelium; the ampulla was comprised of a smaller lumen, a developed muscular layer (168.19 ± 5.05 μm), and was lined by a ciliated epithelium with more folds; and the isthmus was comprised of a more developed muscular layer (183 ± 4.68 μm) and a simple non-ciliated columnar epithelium with small projections. 

The uterus consisted of the endometrium, which contained numerous uterine glands, extended into the submucosa, and was lined by a simple columnar epithelium; the myometrium, which consisted of an inner circular and outer longitudinal layer of smooth muscle fibers, with large blood vessels interspersed between these layers; and the perimetrium, which was made up of connective tissue [25,41,42]. De Oliveira et al. [42] further explained that the endometrium was lined by a simple cuboidal epithelium during diestrus and by a simple columnar epithelium with basal nuclei during estrus. The endometrial epithelium was thicker during estrus than during diestrus and the proportion of endometrial glands were also greater during estrus than during proestrus, metestrus, and diestrus. In addition, Martins et al. [43] found that non-nulliparous females generally had a thicker uterine wall and mucous layer.

Singh et al. [41] stated that the cervix’s epithelium was thrown into many folds projecting into the cervical lumen and was lined by simple columnar cells. In addition to the columnar cells, Mayor et al. [25] found that the epithelium was also monostratified, containing many mucous-secreting cells, with pregnant animals and those in the luteal phase of the estrous cycle having a higher number of secretory cells. De Oliveira et al. [42] made similar findings, but expounded by stating that it had a thin middle layer, mainly consisting of a circular muscular layer and the mucosa of the endocervix comprised of a simple columnar epithelium, while the ectocervix was lined with a stratified squamous non-keratinized epithelium. In addition, more PAS-positive secretory cells were found in pregnant animals in comparison to those in diestrus or metestrus.

The tunica mucosa, tunica muscularis, and tunica adventitia made up the vaginal wall [41,42]. The vaginal mucosa was thrown into folds, with the cranial portion being lined by a simple columnar epithelium, which contained goblet cells, and the caudal portion contained a stratified squamous epithelium [41]. The submucosa was extremely vascularized and the muscularis externa consisted of circular and longitudinal layers of smooth muscle fiber walls [41]. Weir [20], however, noted that there was a change in the vaginal epithelium whereby the epithelial cells differentiated into mucous-secreting cells during proestrus and pregnancy. Similarly, Mayor et al. [25] noted that the stratified squamous epithelium of females, in the follicular phase of the estrous cycle, was thicker (88.3 ± 40.6 µm with 8.0 ± 3.9 cell layers) and more cornified than those in the luteal phase (19.8 ± 1.8 µm with 1.8 ± 0.5 cell layers) or pregnant (22.8 ± 1.9 µm with 1.9 ± 0.5 cell layers). De Oliveira et al. [42] also found that the mucosal epithelium varied with the estrous cycle phases. Table 4 shows the predominant cells and any unique characteristics during the different estrous cycle phases.

The spicules, which were found at the top of the clitoris, consisted of a stratified keratinized epithelium above a vascularized loose connective tissue [42]. During the estrogenic phase, the spicules were large and visible, however, during the diestrus phase, they were small and retracted.

### 3.3. Reproductive Physiology

Courtship in the agouti was initiated by short naso-nasal contact and vocalization, followed by repeated enurination (whereby the female was sprayed by the male’s urine, which then caused her to go into a “frenzy dance”), trembling of the front feet (male), and chasing of the female. Mounting would then be attempted and, once successful, intromission and ejaculation followed [47].

Similar to other hystricognaths, the placenta was lobulated and spherical, containing a vast network of blood vessels for countercurrent exchange, with the characteristic vascular sub-placenta, which is unique to hystricomorphic rodents [40,48]. Apart from having this unique sub-placenta, Mayor et al. [25] also stated that it was a discoidal haemochorial placenta. The placenta was a source of steroid hormones [28]. Though poorly understood, the sub-placenta was said to produce hormones that may be directed towards fetal circulation [49].

Blood collection for hormonal analysis (serum estradiol and progesterone) could also aid in monitoring the estrous cycle [50]. This was performed using a 1 mL syringe to obtain blood via the cephalic or saphenous vein, which was placed in glass tubes and centrifuged in preparation for analysis via amplified electrochemiluminescence. The animals that showed external signs of estrus had corresponding increases in their estrogen levels (22.4 pg/mL and 17.5 pg/mL) and low progesterone levels (3.5 ± 0.2 ng/mL) under the cloprostenol only method of estrus induction [50]. When cloprostenol was used in combination with a gonadotropin releasing hormone (GnRH) analogue, those that had external signs of estrus had corresponding increases in their estrogen levels (12.3 pg/mL, 8.5 pg/mL and 8.3 pg/mL) and low progesterone levels (2.5 ng/mL and 2.3 ng/mL) [50]. Two peak periods of 17β-estradiol were also observed: one during metestrus and the other during proestrus [51,52]. Table 5 shows the average progesterone and 17β-estradiol plasma levels during the estrous cycle. 

Blood collection for hormonal analysis could also be used to monitor pregnancy and not only the estrous cycle phases [53]. Plasma progesterone, which was measured by radioimmunoassay from *D. prymnolopha*, showed an increase of progesterone until week 6 of gestation and then a decline until parturition, with the agouti having post-partum oestrus at 12 days [53].

### 3.4. Reproductive Technologies

One method of monitoring the estrous cycle was by vaginal cytology [18,19,22,41,50]. A moistened (saline solution) swab was introduced into the vagina, where it was used to rub against the vaginal wall before being placed on a glass slide. After drying at room temperature, the smears were stained with rapid panoptic and observed under a light microscope [22,50]. Other authors utilized a 10% saline solution vaginal wash, whereby 1 mL of the final wash was used to make a smear on the slide that was viewed under the microscope, confirming the presence of different cells according to the estrous cycle phase [54]. In proestrus and estrus, superficial cells predominated, while intermediate cells predominated during metestrus and parabasal cells in diestrus [19,22]. External estrus signs coincided with the predominance of superficial cells [22,50].

Another method of monitoring the estrous cycle was via ultrasonography [22,50]. Real time trans-abdominal ultrasound was used to monitor the ovaries, however, no differences in the ovaries could be noted during the different estrous cycle phases [21,22]. Peixoto et al. [50] noted that only one follicle (0.23 cm) in one female appeared round with a hypoechoic center. Similarly, while ovarian follicles were seen as round, hypoechoic structures (averaging 1 mm in diameter), the corpus luteum appeared hyperechoic, averaging 1.4 mm in diameter [22]. External estrus signs coincided with the development of ovarian follicles [22,50].

Estrus was induced using two different protocols, which also involved the use of anaesthesia for restraint (ketamine at 15 mg/kg, intramuscularly, and xylazine at 1 mg/kg, intramuscularly) [50]. The first protocol involved the administration of cloprostenol (5 µg) intraperitoneally, nine days apart. The second protocol was the intravulvar administration of a GnRH analogue (30 µg), followed seven days later by cloprostenol (5 µg) intraperitoneally and another GnRH analogue dose two days later. The first protocol resulted in only 40% of the animals being induced, with estrogen peaks at three and six days after administration of the second dose of cloprostenol [50]. The second protocol resulted in 40% of the animals showing estrus four days after the second dose of the GnRH analogue was administered, while 20% showed estrus ten days after the second dose of the GnRH analogue was administered [50]. However, the authors stated that these protocols had limited efficiency in estrus induction of animals in the luteal phase of the estrous cycle and, even though it was induced, there was no synchronization [50]. 

The first ovarian tissue xenograft of fresh and vitrified fragments was successful in promoting the return of ovarian activity in SCID mice [55]. The authors used ovarian fragments taken from agoutis and placed some into SCID mice, while some were vitrified on a solid surface using a solution consisting of minimum essential media (MEM) plus fetal bovine serum, 0.25 M sucrose, an association of 3 M dimethylsulfoxide (DMSO) with 3 M ethylene glycol (EG), and finally, transplantation to the recipients. Resumption of ovarian activity occurred 21 days after xenografting (in 80% that received fresh ovarian tissue and 16% that received vitrified tissue) and was characterized by the presence of typical signs of proestrus and estrus.

Cryopreservation of ovarian tissue was explored using different cryoprotectant agents and their ability to preserve preantral follicles morphology and the ultrastructure was evaluated [56]. This was done by placing ovarian fragments (for 10 min) in a solution containing MEM plus fetal bovine serum and 1.5 M of one of the three cryoprotectants (DMSO, EG, or propanediol [PROH]), followed by cryopreservation in a programmable freezer. After exposure and/or thawing, the samples were fixed in Carnoy for histological analysis and in Karnovsky for ultrastructural analysis. Analysis of the results indicated that PROH was the best cryoprotectant agent of the three evaluated and thus was recommended. The use of DMSO and EG for solid-surface vitrification (SSV) was also proven effective in the preservation of ovarian tissue, with 3 M EG being the most effective [16].

Another study evaluated the different methods for ovarian tissue vitrification not only on the preservation of preantral follicles, but also on the microbiological load, whereby the fragments of the fresh control and vitrified culture groups were assessed for bacteria and fungi [57]. Ovaries were collected after slaughter and processed. Fragments from each pair of ovaries were allocated into different groups (fresh control, cultured control, SSV control, ovarian tissue cryosystem [OTC] control, cultured SSV, and cultured OTC groups). Ovarian tissue vitrification was done by placing fragments in a solution containing MEM plus fetal calf serum, 0.25 M sucrose, and 3.0 M EG for both SSV and OTC methods. After storage for two weeks and then re-warming, the fragments underwent in vitro culture for 24 h. Following analysis, it was concluded that the methods evaluated were efficient (70%) in preserving ovarian tissue, preantral follicle morphology, and DNA integrity, with the OTC method providing better conditions to prevent bacterial proliferation [57].

The ovarian slicing technique was conducted in order to obtain oocytes from *D*. *prymnolopha* [58]. Ovaries were obtained via ovariohysterectomies of anaesthetized animals. They were subsequently sliced and, using a stereomicroscope, cumulus-oocyte complexes were identified and quantified. Subsequently, they were placed in maintenance media and the morphology was assessed. This method of obtaining oocytes proved to be successful in the collection of a large number of cumulus-oocyte complexes with different degrees of quality [58].

## 4. Capybara (*Hydrochoerus hydrochaeris*)

### 4.1. Gross Anatomy of the Female Reproductive Tract

The ovaries, which were symmetrical and oval in shape, measured 2.6 cm (length) on average and had an ovarian bursa and the associated ovarian and utero-ovarian arteries and ligaments [59,60]. The paired fallopian tubes were supported by the mesosalpinx and consisted of the papilla, isthmus (right length–8.4 cm; left length–9.0 cm), ampulla (right length–4.1 cm; left length–4.2 cm), and infundibulum [60]. The muscular and mucosal layers were supplied by a plexus originating from the ovarian and cranial uterine arteries. Fimbriae surrounded the free margin of the infundibulum, while the inner surface contained wide folds that converged to form the abdominal ostium (a small opening), which led into the ampulla. The ampulla had the largest lumen and folded walls, which decreased in diameter and degree of folds as it moved along to form the longest and narrowest part, the isthmus. The opening of the isthmus into the uterine horns consisted of evaginated projections, the tubal papillae fimbriae [60].

Two broad uterine horns entered into a duplex uterus [59,60,61]. The right uterus measured 12.5 cm in length while the left measured 11.8 cm in length [60], which was similar to the findings by Kanashiro [61]. The uterine horns were supported by the mesometrium and intercorneal ligament [60,61]. Capybaras were said to have a relatively short (4.4 cm) double cervix that projected into the vaginal cavity (external uterine ostia). One was circular in shape, whereas the other was semi-lunar in shape, however both had longitudinal striations and they both secreted mucus [60]. 

The vagina was described as a tubular organ with an average length of 14.4 cm and hyperpigmentation unique to each individual was noted [60]. The external cervical ostia was located cranially, the vaginal vestibule caudally, the rectum dorsally, the pelvic walls laterally, and the urinary vesicle and urethra ventrally. The vulva, anus, and two sinuses were found in a urogenital sac, with the clitoris located within a fossa in the ventral commissure of the vulva [60].

### 4.2. Histology of the Female Reproductive Tract

The ovary was covered by a simple cuboidal epithelium, except in the ovarian hilum area, which had a highly vascularized stroma that contained follicles at different developmental stages and the corpus luteum [60]. The infundibulum had a mucous layer, containing folds covered by a ciliated, pseudostratified epithelium [60]. Ciliated cylindrical cells, non-ciliated cylindrical cells, and cells containing clear cytoplasm and a small central nucleus were present within this epithelium. The ampulla had similar histological findings, with the exceptions being that some small areas of the epithelium were covered by a simple cylindrical epithelium and the serosa and muscular layers were thin [60]. The mucous layer of the isthmus contained only ciliated and non-ciliated cylindrical cells within the pseudostratified epithelium, which had few cilia and some areas covered by a simple cylindrical epithelium [60]. 

The uterus was composed of three tunics; mucosa, muscular, and serosa [61]. The endometrium was lined by a pseudostratified epithelium [60,61]. The lamina propria had diffuse lymphocytes and deeper areas had dense connective tissue with tubular glands, which were covered by a simple cylindrical epithelium [60]. The inner circular layer of the myometrium was thicker than the longitudinal layer and contained smooth muscle fibres and lymphocytes. The perimetrium was only made up of one or two rows of fibroblasts and mesothelium [60]. 

The double cervix contained longitudinal folds covered by a cylindrical stratified epithelium with cuboidal basal cells and superficial polymorphic cells that had a secretory role [60]. The lamina propria invaginated, forming structures that had lumens containing mucus and cell aggregates. The epithelial cells found in the vagina, via vaginal swabbing, were the parabasal, intermediate, nucleated superficial, and anucleated superficial cells [62].

### 4.3. Reproductive Physiology

Females in estrus were detected by behavioural changes in the males, including increased displays and vocalization [33]. The dominant male would keenly follow a female in estrus, sniffing her vulva. The pair would then enter the water, however the female would return to land and re-enter the water allowing copulation to take place, thus mating occurred in the water. Subordinate males were usually chased away by the dominant one [54].

Hormonal evaluation was done in conjunction with vaginal cytology to determine the relationship between the plasmatic estradiol to progesterone (E2/P4) ratio, the maturation index (MI), and the karyopyknotic index (KI); the latter two indices were based on vaginal cytological examination [63]. Blood was collected via heparinized vacutainers to obtain plasma and then estradiol and progesterone were assayed using solid-phase radioimmunoassay (RIA). The plasma results were then compared to the MI and KI. Median plasmatic values of estradiol and progesterone were 326.49 ± 202.61 pg/mL and 121.61 ± 59.17 ng/mL, respectively, and the intra-assay coefficient of variation was 11%. The author, however, did not obtain the expected results as no positive correlation was noted between the high estradiol levels and vaginal cytology [63]. Also, the placentation is characteristic of hystricomorphic rodents, i.e., contains a sub-placenta [64].

### 4.4. Reproductive Technologies

Colpocytology was performed in order to describe the vaginal cell types present during a 30-day period [62,63]. Vaginal swabbing was performed using a moistened swab (saline solution) introduced into the dorsal vaginal commissure. The slides were fixed and stained using Giemsa (2%), Methylene Blue (2%), or Shorr stains for analysis using light microscopy [62]. It was shown that Giemsa (2%) and Methylene Blue (2%) were more practical to use. Barbosa et al. [62] found four types of cells, as well as erythrocytes, leukocytes, and bacteria. However, de Miranda [63] classified cells, in order from smallest to largest, as basal, parabasal, small intermediate, medium intermediate, large intermediate, superficial, and anucleate. The author noted that an increase in cell size corresponded to an increase in acidity and decrease in the nucleus–cytoplasm ratio. 

## 5. Paca (*Cuniculus paca*)

### 5.1. Gross Anatomy of the Female Reproductive Tract

The ovaries were yellowish and oval in appearance, with a smooth surface that had small transparent areas [65,66]. Matamoros [65] found that they measured, on average, 0.8 cm in length and 0.5 cm in width, with further biometric data available in Table 6. Feliciano et al. [66] also noted the presence of follicles within the ovaries based on echogenicity during ultrasonography. The ovaries were said to be supported by an incomplete bursa ovarica, which was attached by the mesovarium [67,68]. 

Continuous with the medial surface of the ovaries was the fallopian tube, a paired narrow, hollow organ [65,67]. Matamoros [65] found that it measured 5 cm in length, which was similar in comparison to the findings of Mayor et al. [68], who stated that the total length was 5.10 cm in non-pregnant females in the follicular phase. Non-pregnant females in the luteal phase had a total length of 4.27 cm, while it measured 5.29 cm in pregnant females. However, Reis et al. [67] found shorter fallopian tubes (right and left—3.69 cm) with diameters of 0.11 cm (right and left).

The uterine horns were interconnected by the intercornual ligament and attached to the abdominal wall via the mesometrium [67]. They were found to be longer in pregnant females, having an average length of 19.51 cm, as opposed to 12.70 cm in non-pregnant females in the follicular phase and 13.97 cm in non-pregnant females in the luteal phase [68]. A much shorter length of 9.63 cm, on average, with a diameter of 2.60 cm was found by other authors [67]. Reis et al. [67] noted that they converged caudally to form a septum, which created two cervical canals with two internal uterine ostia and one external uterine ostium.

The uterus was located in the sublumbar region, which is dorsal to the urinary bladder [67]. It was found to be bicornuate, with each horn measuring about 12 cm in length [65]. The false body was formed by an adjoining thin membrane. In contrast, Mayor et al. [68] described the uterus as being duplex, as both uteri had separate cervices that opened into the vagina. The cervix was found to be longer and wider in non-pregnant females in the follicular phase, having an average length of 2.08 cm and diameter of 2.12 cm, as compared to 1.43 cm (length) and 1.60 cm (diameter) in non-pregnant females in the luteal phase and 1.70 cm (length) and 1.89 cm (diameter) in pregnant females [68].

The vagina, measuring 14 cm in length, could be found in the pelvic region, dorsal to the urinary bladder and ventral to the rectum [65,67]. The external opening, the vaginal orifice, could be completely closed in some females [65]. Also present was the clitoris. Mayor et al. [68], however, found different lengths; 9.23 cm (non-pregnant females in the follicular phase), 10.78 cm (non-pregnant females in the luteal phase), and 9.84 cm (pregnant females). The urethra opened independent of the vagina, next to the clitoris [67]. The vulva was flat and located below a ventral anal depression, below which the clitoris was found [67]. The clitoris was conical in shape and covered by skin at the apex, which revealed two spines when retracted [45]. Females also had one pair of axillary and one pair of inguinal mammary glands [65].

### 5.2. Histology of the Female Reproductive Tract

The ovaries of a 2-month old female did not have a well-defined medulla, as the cortex predominated [65]. The outer part of the cortex contained numerous primary follicles, while the inner part had numerous growing follicles. The ovaries of a pregnant adult, however, differed in the number of follicles and corpora lutea [65]. The ovary of the pregnant horn had the main corpus luteum, three accessory corpus luteum, and Graafian follicles. The ovary of the non-pregnant horn contained six follicles and eighteen accessory corpus luteum. Pregnant females were found to have a greater number of accessory corpus luteum and antral follicles as compared to females in the luteal phase [68]. The mucosal surface of the oviduct contained numerous folds that were lined by a simple columnar epithelium [65].

The endometrium of the pregnant female contained developed uterine glands (in the non-pregnant body), which had a cylindrical epithelium, thin muscular layer, and a serous layer [65]. In the pregnant body, there was an embryo attached via a discoid (hemochorial-type) placenta. The two-month-old female also had numerous developed uterine glands. The muscular layer was thin, made up of an inner circular layer and outer longitudinal layer. The pregnant uterine horn was also larger in diameter, with a thinner endometrium and myometrium in the location of the placenta [68]. They also found the same type of placenta as Matamoros [65], with a sub-placenta located in only one part of the uterus. 

The vagina of non-pregnant females in the follicular phase contained developed stratification and cornification; while non-pregnant females in the luteal phase showed non-developed stratification and pregnant females had columnar secretory cells apically [68]. The axillary mammary glands of pregnant animals contained developed alveoli, with lobules that were not well differentiated but contained fine lines of interlobular connective tissue [65]. The inguinal mammary glands of pregnant animals contained developed alveoli with lobules that were well differentiated and consisted of connective tissue and striated muscle fibres [65]. 

### 5.3. Reproductive Physiology

The courtship behaviour was similar to that of the agouti, except that the female pacas would attack by growling and made attempts to bite. The male pacas were also not as vocal nor did trembling of the front feet occur during courtship—as was the case in the male agouti [47]. Evaluation of fecal progesterone and estrogen, during the estrous cycle and pregnancy, was used to aid pregnancy diagnosis [69]. The results obtained showed that plastic beads were successful as fecal markers and that pregnancy diagnosis could be confirmed via fecal estrogen, but not fecal progesterone. In non-pregnant females throughout the estrous cycle, fecal progesterone levels varied between 0.37–7.9 ng/g dry feces, while fecal estrogen levels varied between 5.08–37.72 ng/g dry feces, with no sustained progesterone peak characteristic of the luteal phaseor any cyclic estrogen peaks observed [69]. In pregnant females, fecal progesterone levels varied between 1.33–6.42 ng/g dry feces, while fecal estrogen levels varied between 8.97–1964 ng/g dry feces [69]. Similar to the other hystricomorphic rodents, the placenta was lobulated with a sub-placenta [70].

### 5.4. Reproductive Technologies

Four phases of the estrous cycle were identified using colpocytology [37,71]. Vaginal swabbing was done and smears were stained using the Harris–Shorr technique, followed by evaluation using a light microscope [71]. In proestrus, intermediate, parabasal, and superficial cells, along with leucocytes, were observed [71]. Similar findings were made by Guimarães et al. [37], who also noted a progressive increase in the number of nucleated superficial cells and a decrease in the other types of cells—with this phase lasting 7–12 days. During estrus, nucleated superficial cells were predominant, which aggregated and became anucleated superficial cells towards the end of the phase [71]. Contrary to other authors, estrus was predominated by anucleated superficial cells and no mucus was seen, even though clinically there was an increase in vaginal mucus and this phase lasted 1.05 days [37]. During metestrus, there were large amounts of intermediate, parabasal, metestrum, and foam cells, as well as leucocytes, and less superficial cells were observed [71]. These results agreed with Guimarães et al. [37], with this phase lasting 4–9 days. During anestrus, there was a prevalence of parabasal cells and debris and a small number of intermediate and metestrum cells and leucocytes [71]. Guimarães et al. [37] did not describe an anestrus phase, rather a diestrus phase, which lasted 7–20 days and was characterized by high numbers of parabasal and basal cells and less leucocytes, with intermediate and other cells in a degenerative process. The mean estrous cycle length was found to be 33.4 days [71], similar to that found by Guimarães et al. [37]. 

Colpocytology was used to identify the different phases of the estrous cycle by another author, however, despite cytological changes, it was not successful in detecting a complete estrous cycle and this was attributed to the handling of the animals [72]. In proestrus, polymorphonuclear intermediate cells and partially keratinized cells were seen. The presence of large, superficial keratinized cells indicated estrus, while intermediate cells and numerous neutrophils indicated diestrus. Also, no anestrus phase was observed [72].

Vaginal cytology, progesterone concentration, and fetal measurements were evaluated during different gestational periods [73]. Vaginal swabbing was done and smears were stained using Shorr and Periodic acid-Schiff techniques, followed by evaluation using a light microscope [73]. At day 30 of pregnancy, less than half the smears had surface cells with estrogenic features and all females had a vaginal discharge. At days 60 and 90, parabasal, intermediate, superficial, and navicular cells were present, with 70% of females at 60 days pregnancy having a vaginal discharge. Despite the gestational period, about half the females had an open vaginal vestibule [73]. Blood for plasma progesterone testing was obtained via venipuncture of the cephalic vein or lateral or medial saphenous veins and evaluated using the radioimmunoassay technique. However, these results were negligible as the majority of animals at days 60 and 90 of pregnancy had less than 1 ng/mL [73]. A 7.5 MHZ convex transducer was used to perform ultrasonography in order to measure the fetal biparietal diameter, with averages of 1.25 cm and 2.34 cm at 60- and 90-days gestation, respectively [73].

Estrus synchronization utilizing progestogen subcutaneous implants, in association with prostaglandin and equine chorionic gonadotropin (eCG) injections, was successfully achieved [74]. The average progesterone concentrations before progesterone implantation were 1.69 ng/mL in proestrus, 0.31 ng/mL in estrus, 1.62 ng/mL in metestrus, and 0.88 ng/mL in diestrus [74]. Two treatment groups were synchronized by implanting 1.5 mg norgestomet subcutaneously, followed seven days later by an intramuscular injection of 0.13 mg of prostaglandin. The implants were subsequently removed 24 h after and the animals were given 25 IU and 50 IU of eCG intramuscularly, respectively [74]. It was noted that only when the implant was removed did the animals go into estrus, with cytology showing intermediate cells with nucleoprotein pigments and intermediate cells of a pre-ovulatory type. The average progesterone value, when the implant was removed, was 1.35 ng/mL [74]. The groups undergoing the treatments, when compared to the control group, had higher pregnancy rates [74]. 

## 6. Comparison amongst *Hystricomorphic rodents*

All three species showed the potential to breed throughout the year, with the capybara showing a seasonal peak in births, based on geographical location. The agouti and paca were medium-sized animals and had fairly long estrous cycles as compared to the much larger capybara. The capybara and paca, however, had similar gestation lengths, with the agouti having a shorter one. They all showed unique placentation by having a lobulated placenta, described as discoidal and haemochorial, with a vascular sub-placenta. All three species gave birth to precocial young.

The agouti was said to have no ovarian bursa, while the lappe had an incomplete one and the capybara had an ovarian bursa. The consensus was that they all had a duplex uterus with a double cervix, with one author stating that the agouti had a partially double uterus and one author stating that the paca had a bicornuate uterus. The capybara had tubal papillae, which were evaginated projections where the isthmus opened into the uterine horns. The vagina of the capybara had hyperpigmentation unique to each individual, which was not noted for the other two species. They also had a urogenital sac containing the eternal genitalia and anus. However, all three had a vaginal closure membrane. The agouti and paca had two spines located on the top of the clitoris, which were visible when the skin was retracted. Histologically, apart from the main corpus luteum, the agouti and paca had many smaller, accessory ones.

## 7. Conclusions

The anatomy of the female reproductive system of all three neo-tropical hystricomorphic rodents showed many similarities, which were consistent with the typical mammalian female reproductive tract. A unique feature of hystricomorphic rodents was the presence of a sub-placenta. They all had relatively long gestational periods, resulting in the birth of precocious young. There were few differences noted and characteristics unique to each species, such as the presence or absence of the ovarian bursa, type of uterus, courting behaviours, and number of mammary glands. All three species had vaginal closure membranes, which could aid in differentiating the phases of the estrous cycle. This thin membrane that covered the vaginal orifice was reformed at each cycle and was present during pregnancy. As a result of this membrane, nothing would be able to exit or enter the vagina unless it is perforated. Further studies on the function of this membrane in these three species would be useful as it relates to their reproduction. A few reproductive techniques were utilized, but it was clear that more work is also needed in this regard, especially since not enough is known about identifying estrus in these animals.

## Figures and Tables

**Table 1 animals-12-00618-t001:** Average morphometric data of the agouti’s ovaries.

	Weight (g)	Length (cm)	Width (cm)	Thickness (cm)	Volume (cm^3^)	Reference
Right ovary	0.08	0.83	0.49	0.24	0.12	[40]
	0.09	0.90	0.53	0.29	-	[42]
	-	1.04	-	-	-	[14]
Left ovary	0.06	0.74	0.45	0.23	0.08	[40]
	0.09	0.92	0.50	0.31	-	[42]
	-	1.05	-	-	-	[14]
Ovary	-	1.04	-	-	-	[41]

**Table 2 animals-12-00618-t002:** Average measurements of the agouti’s (*D. fuliginosa*) vagina, according to Mayor et al. [25].

	Diameter (cm)	Perimeter (cm)
Non-pregnant (follicular phase)	1.63	4.97
Non-pregnant (luteal phase)	1.45	4.44
Pregnant	1.69	4.96

**Table 3 animals-12-00618-t003:** Average morphometric data of the agouti’s ovarian components.

	Diameter (µ)	Reference
Follicle (normal)	800	[20]
Graafian follicle	1000	
Corpus luteum	3000	
Corpus luteum (luteal phase)	4340	[25]
Corpus luteum (pregnant)	5060	
Accessory corpora lutea	<400 *	[20]
Accessory corpora lutea (follicular phase)	1480	[25]
Accessory corpora lutea (luteal phase)	1230	
Accessory corpora lutea (pregnant)	1260	

* The largest accessory corpus luteum seen was 800 µ.

**Table 4 animals-12-00618-t004:** Vaginal cytology during the different estrous cycle phases, according to de Oliveira et al. [42].

Phase	Predominant Cell Types	Prevalence (%)
Proestrus	Anucleated superficial cells	40.20
	Large intermediate cells	30.20
	Nucleated superficial cells	7.20
Estrus	Anucleated superficial cells	53.00
	Nucleated superficial cells	14.60
Metestrus	Parabasal cells	53.40
	Intermediate cells	25.00
	Anucleated superficial cells	18.40
	Nucleated superficial cells	3.20
	Neutrophils	-
	Vaginal microbiota	-
Diestrus	Small and large intermediate cells	40.60
	Grouped parabasal cells	38.60
	Neutrophils	-
	Large amount of mucus	-

**Table 5 animals-12-00618-t005:** Average progesterone and 17β-estradiol levels during the estrous cycle of *D. prymnolopha*, according to Guimarães et al. [51].

	Proestrus	Estrus	Metestrus	Diestrus
Progesterone (ng/mL)	0.78	2.83	1.49	3.71
17β-estradiol (pg/mL)	2030.98	1910.56	1724.83	1939.94

**Table 6 animals-12-00618-t006:** Average morphometric data of the pacas’ ovaries.

	Length (cm)	Width (cm)	Weight (g)	Reference
Left ovary	1.48	0.85	-	[66]
	2.02	1.38	1.68	[67]
Right ovary	1.29	0.80	-	[66]
	2.03	1.39	1.70	[67]

## Data Availability

Data supporting the results can be found within the manuscript.

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
