# Peer review of "Reproductive Technologies Used in Female Neo-Tropical Hystricomorphic Rodents"

_animals, 2022, doi:10.3390/ani12050618_

Round 1
Reviewer 1 Report
This contribution is novel as it represents the second of two literature reviews focused on the female reproductive anatomy and reproductive technologies used in some Neo-tropical hystricomorphic rodents such as the agouti, the capybara and the paca. This literature review documents the anatomy and histology of the female reproductive systems and the reproductive technologies reported for the mentioned rodents, which is important to use assisted reproductive techniques. The work is well written and organized, complete and appropriate in length. For all these reasons, I recommend the paper for publication in Animals.
Author Response
Thank you for your comments. Revisions were made (not based on your recommendations)

Reviewer 2 Report
Dear Authors,
In my opinion this review manuscript is well prepared and comprehensively describes the features of the female reproductive anatomy and reproductive technologies used in selected Neo-tropical hystricomorphic rodents.
I suggest to correct some misspellings within this manuscript, and also the conclusion should better summarize this review article.
Author Response
Corrections were made based on your comments with respect to the conclusion as well as grammatical corrections in text. Please see attached document with further comments

Reviewer 3 Report
The aim of this review was to report the anatomy and histology of the reproductive tracts of female agouti (D.leporina), capybara (H. hydrochaeris) and paca (C.paca). Reproductive technologies carried out in these species were also described. The topic is interesting as these rodents could serve as protein sources, although there are several aspects that should be finetuned before consideration for publication.
Specific comments
Detailed female reproductive physiology (estrous behavior, hormones, kind of ovulation, mating, endocrine production of the placenta, parturition, etc) whenever available should be included. It will greatly contribute future technologies development.
Avoid anatomical and histological features that are common to all mammals. Also please consider summarizing detailed histological features that would hardly be useful for practice purposes.
The objective should be expressed in past tense
143-145: it is not clear the role and changes of the vaginal membrane in this and the others pecies.
Table 4: cells from where?
252-253: this evidences a technical limitation. So add: COULD be noted….
259 to 279: The physiological part should be above in the description of each species .
280-291: fertility tests have not been carried out so be caution to interpret forced follicular development for fertile estrus of a normal estrous cycle.
313: but also on the microbiological load??? what does this mean?
Table 5: This is usual description of vaginal cytology used by that author, it does not help understanding of cycle. Delete and just add one or two sentences in the corresponding paragraph
401-401: hormonal determinations are not reproductive technologies but just physiology description. Thus they should be detailed above in corresponding paragraph. Same for the other species
Add mammary gland information in the 3 species
507-513: this is reproductive endocrinology
535: do not use capitals for norgestomet
544-545: take care as this may change with geographical localization
563-565: this is common to all mammals.
English should be improved as there are some mistakes
Finally, he addition of pictures would be of great value.
Author Response
Good day Reviewer,
All comments were taken into account and revisions made to text. Please see attached document showing in detail changes made to the document.

Round 2
Reviewer 3 Report
Introduction is now extremely long. Introduction should not be more than 1/3 of a whole MS
Reproductive physiology should be placed as a separate paragraph after anatomy and before technologies in each case.
Mammary glands are anatomy. Please replace
79-80: this is biotech not physiology
103-104: add that you are referring to vaginal cytology
115- 116: stage of pregnancy? Concentrations?
Table 1: should also be in the Physiology section. Introductions do not include tables
185- 193 (Also in the conclusion): Considering this IS a typical anatomical organ of rodents a deeper research should be done and its function/s clarified. There are abundant descriptions/reports of anatomy /physiology of vaginal membrane in rodents.
Table 5: the format could be improved. May be 3 columns?
564-565: long gestation could justify precocity so why “although”?
570: I would say we did not know enough to identify estrus. All species have (covert) particularities.
Author Response
Introduction is now extremely long. Introduction should not be more than 1/3 of a whole MS
This was adjusted.
Reproductive physiology should be placed as a separate paragraph after anatomy and before technologies in each case.
This was done.
Mammary glands are anatomy. Please replace
The information was replaced in their respective sections.
79-80: this is biotech not physiology
This was part of the estrus induction and evaluating the hormones; in the original document it was placed under ‘reproductive technology’ but it was stated that hormonal evaluation is not this, rather a physiological description, so this was expanded upon and placed in the respective section.
103-104: add that you are referring to vaginal cytology
This was done.
115- 116: stage of pregnancy? Concentrations?
This was addressed when it was moved to the new sub-section of “Reproductive Physiology”
Table 1: should also be in the Physiology section. Introductions do not include tables
Noted, this was adjusted.
185- 193 (Also in the conclusion): Considering this IS a typical anatomical organ of rodents a deeper research should be done and its function/s clarified. There are abundant descriptions/reports of anatomy /physiology of vaginal membrane in rodents.
Additional information on the vaginal closure membrane was included after the first rounds of corrections, as this point was made in your first set of comments. It can be seen in lines #188-193. However, a little more information was subsequently added to the conclusion. I do not know what more is needed as it was not extensively studied in the species the review focuses on, so future work is needed here.
Table 5: the format could be improved. May be 3 columns?
The table was re-formatted (3 columns were placed, as well as re-ordering of the cell types into a descending order per phase).
564-565: long gestation could justify precocity so why “although”?
Yes, this was addressed.
570: I would say we did not know enough to identify estrus. All species have (covert) particularities.
This was fixed.
